# GAIT-prop: A biologically plausible learning rule derived from backpropagation of error

**Nasir Ahmad   Marcel van Gerven   Luca Ambrogioni**
Department of Artificial Intelligence
Donders Institute for Brain, Cognition and Behaviour
Radboud University, Nijmegen, the Netherlands
{n.ahmad,m.vangerven,l.ambrogioni}@donders.ru.nl

## Abstract

Traditional backpropagation of error, though a highly successful algorithm for learning in artificial neural network models, includes features which are biologically implausible for learning in real neural circuits. An alternative called target propagation proposes to solve this implausibility by using a top-down model of neural activity to convert an error at the output of a neural network into layer-wise and plausible 'targets' for every unit. These targets can then be used to produce weight updates for network training. However, thus far, target propagation has been heuristically proposed without demonstrable equivalence to backpropagation. Here, we derive an exact correspondence between backpropagation and a modified form of target propagation (GAIT-prop) where the target is a small perturbation of the forward pass. Specifically, backpropagation and GAIT-prop give identical updates when synaptic weight matrices are orthogonal. In a series of simple computer vision experiments, we show near-identical performance between backpropagation and GAIT-prop with a soft orthogonality-inducing regularizer.

## 1   Introduction

One of the fundamental tenets of modern systems neuroscience is that the brain learns by selectively strengthening and weakening synaptic connections. Much research in theoretical neuroscience was guided by the Hebbian principle of strengthening connections between co-active neurons [1–3]. However, it appears that purely Hebbian learning rules may not be effective at learning complex behavioral tasks. In the last fifteen years, the fields of machine learning and AI have been revolutionized by the large-scale adoption of deep networks trained by backpropagation (BP) [4–6]. Deep networks have been shown to mimic the hierarchy of cortical representations [7–9], suggesting a connection between deep learning and the brain. However, BP has since long been considered as biologically implausible [10], based in part on its use of non-local information at individual synapses which carry out weight updates. How such information could be stored, transmitted and leveraged has been a cause for concern [10, 11].

To overcome these implausible aspects of the BP algorithm, approaches have been proposed to approximate or replace implausible computations with more realistic and plausible elements [12–16]. Alternatively methods which approximate backpropagation through energy-based models have also been proposed [17–20]. Among those methods, contrastive Hebbian learning, and generalized recirculation have been shown to produce BP-equivalent updates under specific regimes [18, 21], though these methods require a, rather artificial, alternation between positive and negative phases in order to compute updates.

Target Propagation (TP) is a simpler and more scalable approach which proposes that the loss function at the output layer be replaced with layer-wise, local target activities for individual neurons [22, 23],

an approach which was also partially investigated some years prior [24]. The principle of TP is to propagate an output target 'backwards' through a network using (learned) inverses of the forward pass. Under a perfect inverse, these layer-wise targets are equivalent to the outputs of hidden layers which would have precisely produced the desired output. In a recent review paper, Lillicrap and co-authors suggested an approach named 'neural gradient representation by activity differences' (NGRAD) [25]. They conjecture that the most plausible implementation of an effective learning rule in the brain would consist of projecting error-based information into layer-wise neural activity. Given this conjecture, they highlight TP as a feasible and promising approach. However, it remains unclear how updates computed by TP relate to BP and the associated gradients which would optimise network performance.

In this paper, we develop a theoretical framework to analyse the link between the TP and the BP weight update rules. In particular, we show that TP and BP have the same local optima in a deep linear network. Furthermore, we show that in deep linear networks the two update rules are identical when the weight matrices are orthogonal. However, standard TP cannot be easily linked to BP in the non-linear case, even under conditions of orthogonality. A connection to BP can be fully restored by introducing incremental targets – targets which are an infinitesimal shift of the forward pass toward a target output. Using this approach we derive the *gradient-adjusted incremental target propagation* algorithm (GAIT-prop), a biologically plausible approach to learning in non-linear networks that is identical to BP under orthogonal weight matrices. Unlike TP, this approach can also be approximated in the equilibrium state of network with constant input and weak feedback coupling, connecting our method to activity recirculation and equilibrium propagation [18, 19]. Furthermore, our approach to local, error-based learning encodes the exact gradient descent information desired for optimal learning within target neural activities in a plausible circuit mechanism [25].

To derive the theoretical relations between BP, TP, and GAIT-prop we make use of invertible networks. While a perfect inverse model is not biologically plausible, it affords rigorous theoretical comparisons between these learning algorithms. We also relax this invertible network assumption by training networks with hidden layers of different widths – a case in which there is information loss through the network but which nonetheless can be trained accurately.

## 2 Background on backpropagation and target propagation

We start by reviewing the basics of BP. Let us consider a feedforward neural network with an input layer and $L$ subsequent layers. We describe the output of any given layer, as

$$y_l = g_l(y_{l-1}) = f(W_l\, y_{l-1})\,, \tag{1}$$

where $y_l$ is the output of the $l$-th layer, $W_l$ is a weight matrix and $f(\cdot)$ is the activation function. We denote the 'pre-activations' $W_l\, y_{l-1}$ as $h_l$ and use $y_0$ to denote the input.

We consider a quadratic loss between the network output $y_L$ and a target output $t_L$. Given an input-target pair $(y_0, t_L)$ we can define a quadratic loss function, $\ell$ as

$$\ell = \frac{1}{2}(y_L - t_L)^2. \tag{2}$$

The corresponding BP weight update is proportional to the gradient of this loss and has the following form:

$$\Delta W_l^{\text{BP}} = -\eta\, A_l \left( \prod_{j=l+1}^{L} W_j^\top A_j \right) (y_L - t_L)\, y_{l-1}^\top\,, \tag{3}$$

where $A_l$ is a diagonal matrix with $f'(h_l)$ in its main diagonal and $\eta$ is a learning rate.

Target propagation (TP) is an arguably more biologically plausible learning approach in which the desired target is propagated through the network by (approximate) inverses of the forward computations [22]. In its simplest form, given an input-target pair $(y_0, t_L)$, standard TP prescribes a layer-wise loss of the following form:

$$\Delta W_l^{\text{TP}} = -\eta\, A_l\, (y_l - t_l)\, y_{l-1}^\top \tag{4}$$

where $t_l$ is a layer-wise target obtained by applying a (potentially approximate) inverse network to the output target $t_L$. An exact inverse can be defined by the following recursive relation:

$$t_{l-1} = g_l^{-1}(t_l) = f^{-1}(W_l^{-1} t_l)\,. \tag{5}$$

The existence of an exact inverse places constraints on the architecture of the network. In particular, weight matrices must be square, and the activation function must be invertible for any real-valued input. However, this does not require that all layers of a network have the same number of units as we explore in the second half of this paper using auxiliary variables.

The simplicity of TP and its reported performance makes it a leading candidate for biologically plausible deep learning. However, TP is a heuristic method that has not been shown to replicate or approximate BP. In the following, we will derive a series of formal connections between BP and TP. Moreover, we will introduce a new TP-like algorithm that can be shown to reduce to BP under specific conditions.

## 3   Relationship between BP and TP in linear networks

In this section we will reformulate the BP updates of a deep linear network in terms of local activity differences. We begin by rewriting the output difference $(y_L - t_L)$ in terms of $l$-th layer activity differences $(y_l - t_l)$, where $t_l$ is a deep target obtained by applying a sequence of layer-wise inversions, as in Eq. 5. We will assume the existence of inverse weight matrices $W_l^{-1}$ such that $W_l W_l^{-1} = W_l^{-1} W_l = I$. This assumption constrains both the weight matrix shapes (they must be square) and implies that these matrices must be invertible (full rank). Using inverse weight matrices, we can rewrite our difference term as

$$
\begin{aligned}
y_L - t_L &= g_L(y_{L-1}) - g_L(t_{L-1}) \\
&= f(W_L \, y_{L-1}) - f(W_L \, t_{L-1})
\end{aligned}
\tag{6}
$$

where the target of the $L-1$-th layer, $t_{L-1}$, is defined as in Eq. 5. Since the network is linear, we can ignore the activation function and collect these two terms into the matrix product $W_L(y_{L-1} - t_{L-1})$. This formula can then be applied recursively to an arbitrary depth, leading to the expression

$$
y_L - t_L = F_l \, (y_l - t_l),
\tag{7}
$$

where $l$ is the index of an arbitrary hidden layer and $F_l$ is defined as $F_l = \prod_{k=l+1}^{L} W_k$. Substituting this formula into the BP update rule of a linear network, we obtain a reformulation of BP in terms of local targets such that

$$
\begin{aligned}
\Delta W_l^{\text{BP}} &= -\eta \, (F_l^\top F_l)(y_l - t_l) \, y_{l-1}^\top \\
&= (F_l^\top F_l)\Delta W_l^{\text{TP}} \, .
\end{aligned}
\tag{8}
$$

Since $F_l^\top F_l$ is full-rank under our assumption of invertibility, this equation implies that $\Delta W_l^{\text{BP}} = 0$ if and only if $\Delta W_l^{\text{TP}} = 0$, meaning that BP and TP have the same fixed points in invertible linear networks. Furthermore, these fixed points have the same stability since $F_l^\top F_l$ is positive definite. Finally, in linear networks TP updates are identical to BP updates when all the weight vectors are orthogonal, where $F_l^\top F_l = I$.

## 4   Incremental target propagation

If we assume that the Euclidean distance between the activations $y_l$ and the targets $t_l$ is sufficiently small, we can derive a linear approximation of Eq. 6 for an arbitrary transfer function and extend the above analysis to non-linear networks. However, during the early stages of training, when network outputs are far from targets, such an assumption would be unreasonable. To overcome this issue, we reformulate target difference in terms of an 'infinitesimal increment':

$$
\begin{aligned}
y_L - t_L &= \gamma_L^{-1}(y_L - ((1 - \gamma_L) \, y_L + \gamma_L \, t_L)) \\
&= \gamma_L^{-1}(y_L - t_L^*)
\end{aligned}
\tag{9}
$$

where $\gamma_L$ is a scalar and we have defined a new 'incremental' target $t_L^* = (1 - \gamma_L) \, y_L + \gamma_L \, t_L$. Assuming that $0 < \gamma_L \ll 1$, this new target is an incremental shift from the current network output $y_L$, towards the target network output $t_L$. If $g_L(\cdot)$ (our network's forward pass) is continuous, for any real-valued $\kappa$ there is a $\gamma_L$ such that $\|y_L - t_L^*\|_2 < \kappa$. Therefore, assuming $g_L(\cdot)$ to be a continuous function, we can approximate the resulting difference with a linear function:

$$
\begin{aligned}
y_L - t_L &= \gamma_L^{-1}(g_L(y_{L-1}) - g_L(t_{L-1}^*)) \\
&= \lim_{\gamma_L \to 0} \gamma_L^{-1}( \, A_L \, W_L \, (y_{L-1} - t_{L-1}^*))
\end{aligned}
\tag{10}
$$

where the approximation error is of the order of $\|y_L - t_L^*\|_2^2$. This procedure can be recursively carried out to describe the difference term at our output layer as a function of activity at a layer of any depth such that

$$y_L - t_L = \lim_{\gamma_{l+1:L} \to 0} \left( \prod_{j=0}^{L-(l+1)} \gamma_{L-j}^{-1} A_{L-j} W_{L-j} \right) (y_l - t_l^*) \qquad (11)$$

with the layer-wise incremental target defined recursively as

$$t_{l-1}^* = g_l^{-1}((1 - \gamma_l) y_l + \gamma_l t_l^*) . \qquad (12)$$

We can now define an incremental TP-based update, $\Delta W_l^{\text{ITP}}$ where

$$\Delta W_l^{\text{ITP}} = -\eta A_l (y_l - t_l^*) y_{l-1}^\top . \qquad (13)$$

Substituting Eq. 13 into the BP update rule (Eq. 3), we obtain an asymptotic linear relation between BP and ITP updates:

$$\Delta W_l^{\text{BP}} = \lim_{\gamma_{l+1:L} \to 0} M(h_l, \ldots, h_L) \Delta W_l^{\text{ITP}} \qquad (14)$$

with

$$M(h_l, \ldots, h_L) = \left( \prod_{j=l+1}^{L} \gamma_j^{-1} \right) A_l \left( \prod_{j=l+1}^{L} W_j^\top A_j \right) \left( \prod_{j=0}^{L-(l+1)} A_{L-j} W_{L-j} \right) A_l^{-1} . \qquad (15)$$

Unfortunately, since $M(h_l, \ldots, h_L)$ depends on the layer-wise activity (via the diagonal matrices of derivatives, $A_l$), Eq. 14 does not imply an equivalence between the fixed points of BP and ITP for a dataset bigger than one sample. Furthermore, orthogonality of weight matrices is also unhelpful due to these same multiplications by layer-wise activation function derivatives. In general, since these derivatives depend upon the input data, it is not possible to find a constraint on the weights that restores the equivalence of the linear case. Fortunately, we can restore this equivalence by further modification of the incremental target, as we demonstrate in the next section.

## 5 Gradient-adjusted incremental target propagation

By incorporating the data-dependent derivatives into the $\gamma_l$ variables, we can recover an equivalence between BP and an approach which makes use of layer-wise targets. Specifically, by introducing

$$\epsilon_l = \gamma_l A_l^2 , \qquad (16)$$

we can define gradient-adjusted incremental targets

$$t_{l-1}^\dagger = g_l^{-1}((1 - \epsilon_l) y_l + \epsilon_l t_l^\dagger) . \qquad (17)$$

Given this new target formulation, we can now define the 'gradient-adjusted incremental target propagation' (GAIT-prop) update

$$\Delta W_l^{\text{GAIT}} = -\eta A_l (y_l - t_l^\dagger) y_{l-1}^\top . \qquad (18)$$

If we express the GAIT-prop-based update in terms of the BP updates we find

$$\Delta W_l^{\text{BP}} = \lim_{\gamma_{l+1:L} \to 0} N(h_l, \ldots, h_L) \Delta W_l^{\text{GAIT}} , \qquad (19)$$

with

$$N(h_l, \ldots, h_L) = \left( \prod_{j=l+1}^{L} \gamma_j^{-1} \right) A_l \left( \prod_{j=l+1}^{L} W_j^\top A_j \right) \left( \prod_{j=0}^{L-(l+1)} A_{L-j}^{-1} W_{L-j} \right) A_l^{-1} . \qquad (20)$$

On inspection, this matrix formulation (aside from the preceding gamma terms) reduces to the identity matrix when all weight matrices $W_l$ are orthogonal matrices. This allows us to express the following equivalence formula

$$\Delta W_l^{\text{BP}} = \lim_{\gamma_{l+1:L} \to 0} \left( \prod_{j=l+1}^{L} \gamma_j^{-1} \right) \Delta W_l^{\text{GAIT}} \qquad (21)$$

as being true under the assumption of orthogonality of weight vectors. In practice, a fixed small value is used in place of the $\gamma$ terms instead of a layer-wise infinitesimal.

Note that the GAIT-prop update rule (Eq. 18) only uses locally-available information and it is in this sense as biologically plausible as the classic TP target. Pseudocode for GAIT-prop is given in Alg. 1.

---
**Algorithm 1** GAIT-prop (per training sample update)
---
**for** $l = 1$ **to** $L$ **do**
   $y_l \leftarrow f\left(W_l\, y_{l-1}\right)$
**end for**
With output target, $t_L^\dagger \leftarrow t_L$
**for** $l = L - 1$ **to** $1$ **do**
   $t_l^\dagger \leftarrow g^{-1}\left((1 - \epsilon_{l+1})\, y_{l+1} + \epsilon_{l+1}\,(t_{l+1}^\dagger)\right)$
**end for**
**for** $l = 1$ **to** $L$ **do**
   $\ell_l(W_l) = \gamma_{L-l}^{-1}\left(y_l - t_l^\dagger\right)^2$
   Update $W_l$ by SGD on the quadratic loss function $\ell_l(W_l)$, treating $t_l^\dagger$ as a constant.
**end for**
---

### GAIT-prop with an arbitrary loss function

Though we assumed a quadratic loss above (and make use of a quadratic loss function for our results section), we can also make use of GAIT-prop in order to compute local layer-wise updates for an arbitrary loss function computed on the output unit activations. In particular, GAIT-prop's key requirement is a formulation of the gradient using a difference between the forward pass activity and some target activity at the output layer (consider Equation 3 for output layer $L$). From such a formulation, layer-wise targets can be computed.

Take an arbitrary loss function $\mathcal{L}(y_l)$, such that our final layer weight updates computed by back propagation can be written:

$$\Delta W_L^{BP} = -\frac{d\mathcal{L}(y_L)}{dW_L} = -\frac{d\mathcal{L}(y_L)}{dy_L}\frac{dy_L}{dW_L}.$$

We can now introduce the forward-pass activity to this formulation without introducing any errors or approximations, such that

$$\Delta W_L^{BP} = -\left(y_L - y_L + \frac{d\mathcal{L}(y_L)}{dy_L}\right)\frac{dy_L}{dW_L} = -(y_L - t_L)\frac{dy_L}{dW_L}.$$

where the target, $t_L$, is defined

$$t_L = y_L - \frac{d\mathcal{L}(y_L)}{dy_L}.$$

Thus, any arbitrary loss function computed on the output units can be written as a difference between forward-pass activations and a constructed target activation of the form provided above. From here, the GAIT-prop derivation provided above holds and we can make use of GAIT-prop for any such loss function.

## 6 GAIT-prop in a neural circuit

One significant weakness in the biological plausibility of TP is that the target signal needs to be propagated backward without being 'contaminated' by the forward pass. This requires either a parallel network for targets alone, a complete resetting of the network (with blocking of inputs), or some sophisticated form of compartmentalized neurons capable of propagating two signals in both directions.

In comparison, the incremental nature of the layer-wise targets produced by GAIT-prop makes it particularly suitable for an implementation in a biologically realistic network model. Figure 1, Left, depicts the differences between both algorithms. The backward-propagated signals for GAIT-prop are (weakly) coupled to the forward pass, meaning that during target presentation both input (forward-pass) and targets are simultaneously presented. In fact, the ITP algorithm (equivalent to GAIT-prop in a linear network) can be shown to emerge in the equilibrium state of a simple dynamical system model of a neural network with feedback connection (see Supplementary Material).

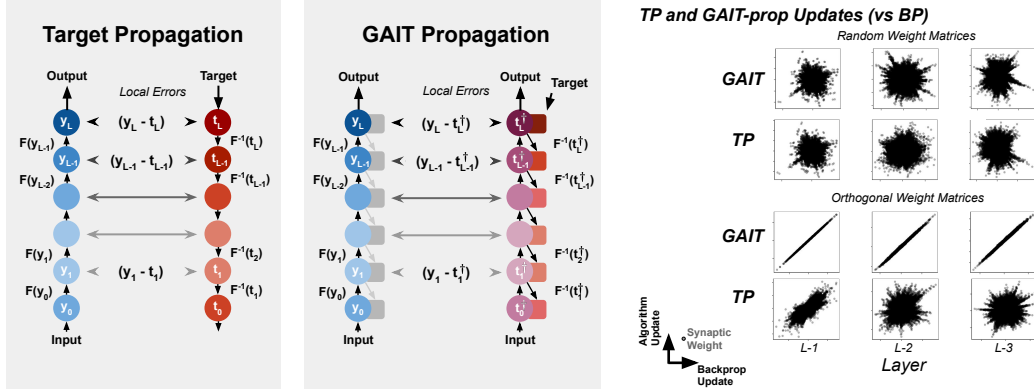

Figure 1: **Left:** Graphical depiction of TP versus GAIT-Prop. **Right:** Scatter plots showing the alignment of TP and GAIT-prop weight updates against BP. These are shown for updates to an untrained (square) network with random or orthogonal weight initializations.

Figure 1, right, shows the efficacy of the coupling proposed in GAIT-prop when combined with orthogonal weight matrices. The weight updates produced by GAIT-prop in this condition almost perfectly equals those computed by BP. Furthermore, this coupling reduces the requirement for two non-interfering information flows by suggesting that the same inputs can be present during the target propagation phase.

# 7    Simulated invertible networks and tasks

**Invertible components**    All of the analyses and simulations presented in this paper require an invertible network model. This requires both an invertible activation function and invertible weight matrices.

Aside from linear networks, we use the leaky-ReLu activation function. The inverses of both the linear and of the leaky-ReLu activation functions are trivial.

To ensure invertibility of weight matrices, we only make use of square matrices and empirically find that by random initialisation these remain full-rank during training (and therefore invertible). The use of square matrices places constraints upon the network architectures from which we can choose. The simplest network architecture of choice is a network with a fixed-width, i.e. every layer of the network has an equivalent number of units as the inputs. The consequence of such invertible, fixed-width networks (of width equivalent to the input) is that there is sufficient information at every layer to reproduce the input pattern. However, the tasks we make use of require only ten output neurons (far fewer than the number of inputs) and to accommodate this, we make use of auxilliary units.

**Auxiliary units and information loss**    So far, we assumed that the layer-wise transformations, $g_l(\cdot)$, are fully invertible functions. This requirement places strong constraints upon the network architecture. Specifically, fully invertible architectures require as many output nodes as there are input nodes at every layer and cannot discard task irrelevant information.

In the TP literature, this problem is addressed using learned pseudo-inverses (autoencoders) which can transform between layers of arbitrary size [23]. However, in practice, a target obtained using pseudo-inverses must represent some prototypical target since not all low-level information can be recovered. This can result in the presence of non-zero error terms despite correct network behaviour.

Bartunov et al. [26] first suggested the use of 'auxiliary output' units – additional units at the output of a network which are provided no error signal and are used to store task-irrelevant features so that diverse targets can be produced for the hidden layers of a network. Without these, the targets provided to hidden layers of all examples of a given class are identical. By the addition of these auxiliary variables, diverse targets (which vary across inputs of the same class) can be produced for each layer, improving network performance.

Here, we make use of such auxiliary outputs for our full-width network models. We also extend this approach and relax the assumption of full invertibility by allowing auxiliary units at every layer of a network. By placing auxiliary units (which have no forward synaptic connections) at arbitrary layers of our network models, we can build variable-width networks.

Despite these auxiliary units, weight matrices between layers must remain square for inversion of the non-auxiliary neuron activations. This means that the number of non-auxiliary neurons in some layer indexed $l - 1$ is equal to the number of units in the subsequent layer indexed $l$. We can therefore describe the $l - 1$-th layer to have $N_{l-1} - N_l$ auxiliary units, with activations $z_{l-1}$, and $N_l$ forward projecting neurons with activations $y_{l-1}$. In a forward model the auxiliary units of a lower layer are ignored such that $y_l = g_l(y_{l-1})$. But in the inverse pass, we make use of an augmented inverse transfer function $\tilde{g}_l^{-1}(y_l)$ which maps the activations of the $l$-th layer to the tuple $(y_{l-1}, z_{l-1})$. Using these variables, we can define the GAIT-prop target as before. That is,

$$t_{l-1}^{\dagger} = \tilde{g}_l^{-1}((1 - \epsilon_l)\, y_l + \epsilon_l\, t_l^{\dagger}). \tag{22}$$

Note that the values of the auxiliary neuron activations, $z_l$, are simply copied from the forward pass. Furthermore, unless there are additions to the cost-function, the weights mapping $y_l$ to $z_{l+1}$ do not change during training since the target of the auxiliary variables is always identical to their forward pass values. We can consider this as a case in which auxiliary neurons simply do not receive feedback connections from the task-relevant neurons.

**Encouraging orthogonality**   One desired feature of networks which we wish to train by GAIT-prop is for weight matrices to be (close to) orthogonal. To encourage orthogonality of the rows of our weight matrices, we make use of a regularizer which can be applied layer-wise. This regularizer can be expressed for the $l$-th weight matrix as

$$\lambda\, ||W_l W_l^{\top}\, (J - I)||_2^2, \tag{23}$$

where $J$ is an all-ones matrix (i.e. all elements equal to 1), $I$ is the identity matrix, and $\lambda$ modulates the strength of the regularizer relative to the task-related error. When this regularizer is applied, weight updates are combined with the task-relevant updates and these are collectively scaled by the learning rate, $\eta$. We find that the use of weak regularization is sufficient to ensure that GAIT-propagation remains performant. This regularizer uses non-local (non-plausible) information to enforce orthogonality, however in the Discussion section we explore plausible neural mechanisms that could achieve a similar result.

**Tasks**   We make use of three image classification datasets: MNIST, Fashion-MNIST, and KMNIST. These datasets all consist of $28 \times 28$ (total 784) pixel input images and a label between 1 and 10 indicating the class. We convert the labels into a one-hot vector and during training the quadratic loss between the network output and this one-hot vector is minimised. In the case of TP and GAIT-prop, the one-hot vector is the output layer target.

**Parameters and learning**   The Adam optimiser was used during training of our neural network models. In order to identify acceptable parameters for each of our learning methods, we ran a grid search for the learning rate $\eta$ and the orthogonal regularizer strength $\lambda$. The highest-performing networks were tested for stability and stable high performing parameters were used. Details of specific parameters used and the grid search outcomes are provided in the Supplementary Material. Code used to produce the results shown in this paper is available at https://github.com/nasiryahm/GAIT-prop.

## 8   Results

Figure 2 presents the accuracy the algorithms we have thus far considered. In particular, we find that GAIT-prop is extremely consistent, with performance indistiguishable from backpropagation. This is true for non-linear networks of various depths (Fig. 2A), when applied to different tasks (Fig. 2B) and for linear networks (Fig. 2C). We find that GAIT-prop is highly robust in general – even capable of training non-linear networks of more than four layers without modification (see Supplementary Material). By comparison, TP suffers from lower accuracy in non-linear networks (Fig. 2A and C) even across a large choice of training parameters (see Supplementary Material). Though TP does show high performance and stability in linear networks (Fig. 2C), as expected from our theoretical analyses.

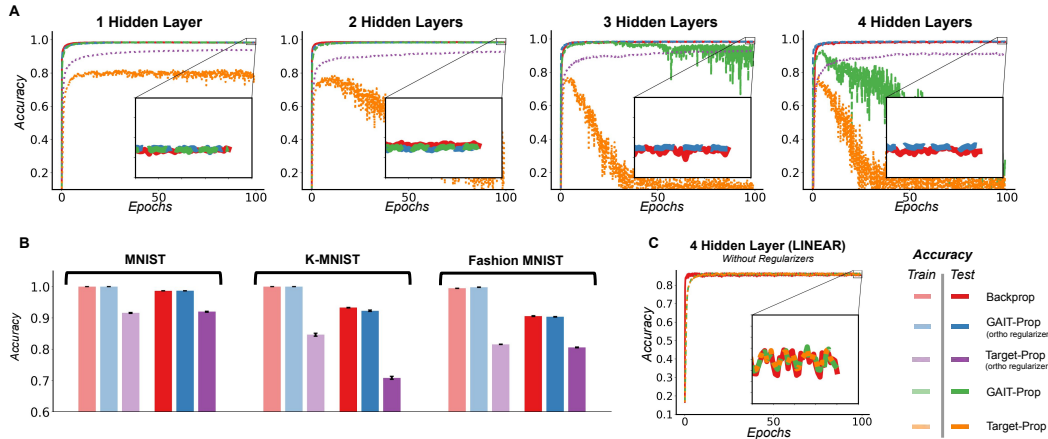

Figure 2: **The performance of multi-layer perceptrons trained by BP, TP, and GAIT-prop.** All results in this figure are in networks with a fixed width network: 784 neurons in every layer. **A:** Test accuracies (MNIST) of the algorithms are compared in non-linear networks of various depth. The networks are trained by parameters as determined by a grid search (see Supplementary Material). **B:** Accuracy of algorithms across tasks (MNIST, KMNIST, and Fashion-MNIST). Non-linear networks with four hidden layers were trained with five repeats (error bars indicate standard deviation). Peak training and test accuracies are presented. **C:** Accuracy of a linear network (with four hidden layers) and no orthogonal regularizers.

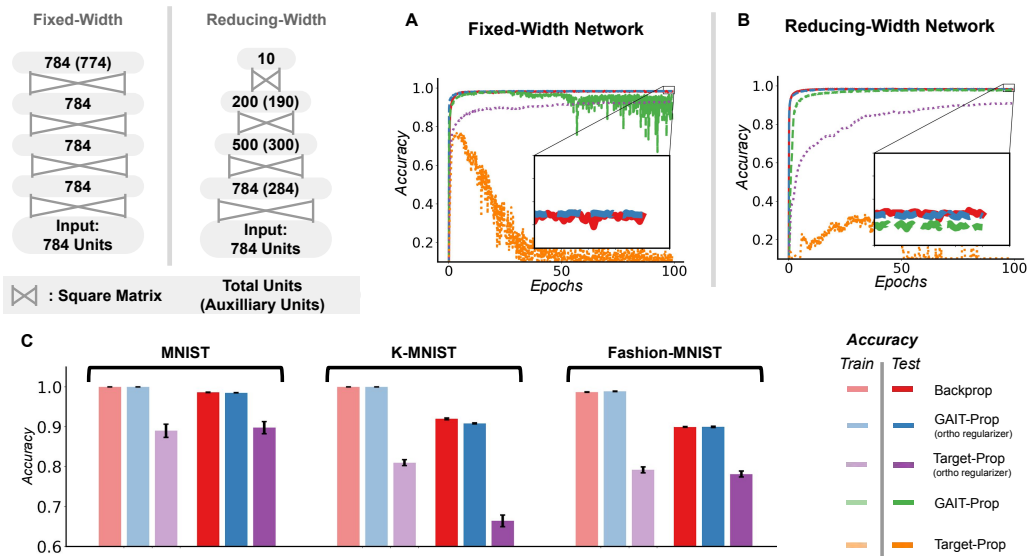

Figure 3: **Performance in non-linear networks with variable hidden-layer sizes. A:** Network performance of a full fixed with network (three hidden layers). **B:** Network performance of a network with reduced hidden layer widths. **C:** Peak accuracy of BP, TP and GAIT-prop in a reducing-width network across datasets and across multiple repeats (error bars indicate standard deviation).

Figure 3 exhibits the performance of networks with variable hidden-layer widths. It can be seen that learning in these networks is as effective for GAIT-prop as it is for backpropagation and the inclusion of layer-wise auxilliary neurons is empirically found to be succesful. Hence, these results show high performance in networks with a relaxed definition of a full inverse.

# 9 Discussion

Our theoretical work and results show that GAIT-prop produces performance indistinguishable from backpropagation. In comparison TP suffers in non-linear neural networks and, as previously described by Bartunov et al. [26], peak performance is highly dependent upon the training parameters (see Supplementary Material). It is possible that with alternative activation functions such as the hyperbolic tangent, TP will show improved performance (as suggested by [26]). However, the $\tanh$ activation function is not invertible for all real values and our analysis of GAIT-prop shows that, unlike TP, its performance is theoretically independent of the employed activation function.

Our proposal of layer-wise incremental variables, $\gamma_l$, may be of interest to readers interested potential biological substrates of such a learning system. These incremental paramters scale down the impact of target (top-down) neural activities to hidden layers. This re-scaling allows a linearisation of the error computation and is therefore a key component of our rigorous mathematical derivation. However, in practice we find that it is also possible to make use of a single incremental variable on the output layer, $\gamma_L$, and carry out inversion without incremental variables for all other layers. The stability of such strong top-down inputs however, is questionable and therefore we consistently make use of weak layer-wise perturbations.

In the results section of this work, we find that a weak orthogonal regularizer is sufficient to enable high and stable performance for GAIT-prop. However, a reader might question how this could arise biologically – this is indeed an open question. We propose that lateral inhibitory learning might aid in sufficient orthogonalization of synaptic weight matrices. As explored previously by King et al. [27], inhibitory plasticity can be used to decorrelate neural outputs. Since weight updates are computed with an outer-product of between-layer neural activities, decorrelated layer-wise activities could encourage some orthogonalization. Since GAIT prop requires a limited strength of orthogonalization, such decorrelating mechanisms may be sufficient – though again this is an open question and avenue for future research. In addition, such simple inhibitory Hebbian plasticity rules have been found to stabilize and balance neural network models, reproducing statistics and observations of cortical activity [28, 29].

One further biologically implausible component of the simulations we have presented are the use of perfect inverse models. In real neural circuits, such an inverse model could be learned by (denoising) auto-encoders, as has been previously attempted with TP [22, 23, 26]. However, the incremental term we have introduced may well become highly sensitive to noise in a network with an imperfect inverse – motivating the use of a relatively large value for this incremental term. Despite this drawback of our current simulation work, our theoretical explorations of the relationship between BP, TP and GAIT-prop required assumption of perfect inverse models and we leave explorations of the non-perfect case to future studies.

In conclusion, we have theoretically and empirically demonstrated that plausible layer-wise targets can be created in a neural network model with (close to) orthogonal weight matrices. The resulting updates lead to learning with almost indistinguishable performance compared to BP. This is accomplished in networks of both fixed and variable layer-widths in a novel application of auxilliary units. Our work elucidates the relationship between BP and local target-based learning and is a significant step forward in the debate surrounding the plausibility of BP for learning in real neural network models.

## Broader impact

This research positively impacts discourse and research into relevant scientific sub-disciplines (including machine learning, computational neuroscience, and neuromorphic computing). Its continued development could lead to algorithms for learning in neuromorphic computing devices, and insights into the nature of credit assignment in biological neural systems. Beyond research and development, this work alone has no broader societal or ethical impact.

## Funding disclosure

The authors declare that there exist no financial or non-financial conflicts of interest pertinent to this study.

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
