[Supplementary Material]

# Supplementary Material
# GAIT-prop: A biologically plausible learning rule derived from backpropagation of error

**Nasir Ahmad   Marcel van Gerven   Luca Ambrogioni**
Department of Artificial Intelligence
Donders Institute for Brain, Cognition and Behaviour
Radboud University, Nijmegen, the Netherlands
{n.ahmad,m.vangerven,l.ambrogioni}@donders.ru.nl

## 1   GAIT-prop as equilibrium states in a linear network

The GAIT-prop and ITP targets are implemented as a weak perturbation of the forward pass. This can be implemented through weak feedback connections. We demonstrate this in the following linear firing rate model:

$$\tau \dot{u}_1(s) = -u_1(s) + x(s) + \nu W^{-1} u_2(s) \, ,$$
$$\tau \dot{u}_2(s) = -u_2(s) + W u_1(s) + t_2(s) \, . \tag{1}$$

where $u_j(s)$ is the firing rate of the $j$-th layer at time $s$, $\nu$ is a small feedback coupling parameter and and $\tau$ is a time constant. Consider a setting when the input $x$ is presented $\delta$ time units before the target $t$ and then stays present throughout the experiment. If $\delta \gg \tau$ and $\nu < 1$, the activity of the first hidden layer, firing rate denoted $u_1(s)$ above, will converge to the an equilibrium-state value, $y_1$, which can be expressed as:

$$y_1 = \left(1 + \frac{\nu}{1-\nu}\right) x = (1+\gamma)x, \tag{2}$$

where $\gamma = \frac{\nu}{1-\nu}$ is our incremental factor. Note that for this incremental factor to remain below 1.0, $\nu < 0.5$. After the target is presented (i.e. $t_2$ is a fixed non-zero value), the firing rate will converge to a shifted steady-state, $\tilde{y}_1$ such that

$$
\begin{aligned}
\tilde{y}_1 &= \left(1 + \frac{\nu}{1-\nu}\right) x + \frac{\nu}{1-\nu} W^{-1} t_2(s) \\
&= (1+\gamma)x + \gamma t_1 \\
&= y_1 + \gamma t_1 \, ,
\end{aligned}
\tag{3}
$$

where $t_1 = W^{-1} t_2$ and represents the inverted target. If we compare the available terms in the two equilibrium states ($y_1$ and $\tilde{y}_1$), these equilibrium points contain sufficient information to make use of the GAIT-prop rule. In particular, the GAIT-prop rule in the linear case (or the ITP rule generally) requires a target of the form $(1-\gamma)y_l + \gamma t_1$. Such a difference term can be trivially computed using these two equilibrium states. Furthermore, the shifted equilibrium state, $\tilde{y}_1$, is as default extremely close to the desired target (the effect of the missing $(1-\gamma)$ factor is not explored).

For non-linear networks, the computation of the full GAIT-prop target also require the computation of the activity dependent term $A_l$. We have not described a particular dynamical system in which this could emerge, however this information is local to each unit and therefore remains biologically plausible.

## 2 Model parameters

The networks we train in this study all made use of the Adam optimiser [1]. Along side the Adam optimiser parameters, we had parameters related to orthogonal regularization and the incremental component of GAIT-prop. All parameters are outline below, many of these were kept fixed and some were tested in a parameter grid search. The table below presents the relevant parameters.

| Parameter | Value |
|---|---|
| Learning Rate of Adam Optimiser | $\{10^{-3}, 10^{-4}, 10^{-5}\}$ |
| $\beta_1$ of Adam Optimiser (fixed) | 0.9 |
| $\beta_2$ of Adam Optimiser (fixed) | 0.99 |
| $\epsilon$ of Adam Optimiser (fixed) | $10^{-8}$ |
| Orthogonal Regularizer Strength ($\lambda$) | $\{0, 10^{-1}, 10^1, 10^3\}$ |
| Incremental Factor for GAIT-prop ($\gamma$, fixed) | $10^{-3}$ |

### 2.1 Learning rate and regularizer grid search

In order to determine favourable parameters for the learning algorithms which we investigated, we ran a grid search over two key parameters: the learning rate, $\eta$, and the strength of the orthogonal regularizer, $\lambda$. This parameter search was carried out in a feed-forward square network with 4 hidden layers and a full output layer of 10 output units and 774 auxilliary units. A leaky-ReLu transfer function was used (as is true for all non-linear network results in this paper).

Note that networks were either initialised with orthogonal weight matrices or by Xavier initialisation [2]. Both in this parameter search and in our main paper, Xavier initialization was used for all networks in which $\lambda = 0.0$. For all non-zero values of $\lambda$, networks were initialised with an orthogonal weight matrix.

The results report peak and final (end of training) accuracy on the training set (organise 'peak / final'). Parameters shown in bold were chosen and used for all results presented in the main paper.

Note that target propagation systematically shows lower accuracy at the end of training compared to at its peak over a large parameter range. We find that target propagation often does best when early-stopping is implemented to 'catch' this peak, unlike the other two algorithms which have asymptotic behaviour. Furthermore, the highest performing parameters for target propagation (indicated in italics) were found to be highly unstable when network depth was modified. This was to an extent that reducing network depth caused a counter-intuitive drop in performance. Therefore, we made use of parameters which had much greater stability in performance across network architectures and structure), shown in bold as for the other algorithms.

Table 1: **Backpropagation**

|  |  | $\lambda$ | | | |
|---|---|---|---|---|---|
|  |  | 0.0 | 0.1 | 10.0 | 1000.0 |
| $\eta$ | 1e-3 | 100.00 / 99.98 | 99.93 / 99.78 | 99.59 / 98.80 | 89.23 / 10.44 |
|  | 1e-4 | **100.00 / 100.00** | 100.00 / 100.00 | 100.00 / 100.00 | 97.08 / 97.04 |
|  | 1e-5 | 100.00 / 100.00 | 99.91 / 99.90 | 99.71 / 99.70 | 96.92 / 96.92 |

Table 2: **Target Propagation**

|  |  | $\lambda$ | | | |
|---|---|---|---|---|---|
|  |  | 0.0 | 0.1 | 10.0 | 1000.0 |
| $\eta$ | 1e-3 | 17.94 / 10.22 | 21.51 / 9.87 | 90.38 / 11.45 | 86.53 / 7.16 |
|  | 1e-4 | 68.11 / 9.74 | 77.5 / 5.57 | 92.02 / 11.10 | 90.53 / 9.38 |
|  | 1e-5 | 77.29 / 9.75 | 82.62 / 13.02 | *93.10 / 92.16* | **91.63 / 90.28** |

Table 3: **GAIT Propagation**

|  |  | $\lambda$ | | | |
|---|---|---|---|---|---|
|  |  | 0.0 | 0.1 | 10.0 | 1000.0 |
| $\eta$ | 1e-3 | 19.34 / 17.94 | 100.0 / 99.91 | 99.74 / 93.79 | 92.44 / 72.24 |
|  | 1e-4 | 93.08 / 26.36 | **100.00 / 100.00** | 99.99 / 99.98 | 97.05 / 96.99 |
|  | 1e-5 | 98.38 / 98.27 | 99.84 / 99.83 | 99.66 / 99.64 | 96.83 / 96.82 |

# 3 Performance of GAIT-propagation in deeper networks

In the main paper, we showed that GAIT-propagation produces networks with final training/test accuracies which are indistinguishable from those produced by backpropagation of error. Those results were shown for networks with up to four hidden layers.

Figure 1: **The performance of deep multi-layer perceptrons trained by BP, and GAIT-prop.** All results in this figure are in networks with a fixed width network: 784 neurons in every layer. Test accuracies (MNIST) of the algorithms are presented here during training in non-linear networks with 6 and 8 hidden layers. The networks use optimal parameters as determined by a grid search (see previous section).

Figure 1 shows that GAIT-prop remains highly performant even in networks with six or eight hidden layers. Performance lags slightly behind that of BP for the eight hidden layer network though it should be expected that in deeper networks our decision to fix the incremental $\gamma$ parameter would lead to a worse approximation of BP (and therefore a decrease in performance). Nonetheless, GAIT-prop remains robust and shows stable training and high performance despite potential increases in approximation errors in deeper networks.

# 4 Code

Code to reproduce the results presented in this work, details of the required python environment, and a README providing some information are provided at

https://github.com/nasiryahm/GAIT-prop.