[Reviews · NeurIPS 2020]

Review 1

Summary and Contributions: This paper leverages the fact that the inverse of an orthogonal matrix is its transpose to to show that a modified version of target propagation approximates backpropagation (assuming invertible layers and orthogonal forward weights). The authors confirm that the proposed algorithm essentially matches the performance of backpropagation in experiments.

Strengths: The connection / equivalence between the proposed method and backpropagation is presented clearly, as is the discussion of auxiliary units and why they are necessary / helpful. The empirical results are strong and confirm the equivalence between GAIT-prop and backprop under ideal orthogonality conditions is robust to a using a soft orthogonality constraint.

Weaknesses: 1. The authors use exact inverses rather than learning approximate inverses with e.g. an autoencoder. Without some mechanism behind it, the use of exact inverses is just as biologically implausible as the use of transpose weights. I think it is important to show that the method is robust to the use of learned inverses. 2. The method hinges crucially on the orthogonality regularizer. Given that this paper's focus is biological plausibility, I think it is important to show or at least propose how this orthogonality condition could be imposed biologically. The authors "hypothesise that lateral inhibitory learning could aid in sufficient orthogonalization of synaptic weight matrices" -- but it is not clear to me that such a mechanism would enforce sufficient orthogonality.

Correctness: I think the experiments have been performed correctly and the claims follow from the experimental results.

Clarity: Yes, it is well written.

Relation to Prior Work: Yes, the related work section discusses the relevant past work on biologically plausible backprop alternatives and discusses how the proposed method deviates from the original target propagation algorithm.

Reproducibility: Yes

Additional Feedback: Update after author response: Thanks to the authors for your comments. I believe this paper makes a meaningful contribution to this line of work and have changed my score accordingly to support acceptance. I do have a few comments that I hope you will consider as you prepare a final version of this paper, mainly coming from a neuroscience perspective. 1. While the method described in this paper advances the family of target prop-related models and may serve as a foundation for future work in bio-plausible learning models, I don't think it is appropriate to describe it as more biologically plausible than backpropagation. One of the commonly cited biologically implausible features of backpropagation (weight symmetry) is replaced here by an equally implausible mechanism (perfect inverse models). It is true that bio-plausible ways of approximating inverses may exist, but there are also proposals for bio-plausible ways of maintaining weight symmetry (e.g. Akrout et al.). On top of that, this method adds another feature, regularization of weight orthogonality, which has no obvious biological implementation. While there may be advantages, from a bio-plausibility perspective, to a target-based / NGRAD framework over error-based backpropagation, this is by no means settled and rather speculative given the current state of experimental evidence in neuroscience. I think this paper would benefit by acknowledging these limitations / uncertainties and framing the method not as a bio-plausible alternative to backprop, but rather as an alternative framework to backprop which could, with further development, yield bio-plausible models. 2. In the paper and author response the authors suggest that there are known biological mechanisms that could serve as a regularizer for weight orthogonality. I don't know of any such mechanisms, and the cited references do not support this point (they seem to be about decorrelation of *activity*, not weights). An orthogonality regularizer is a very nonlocal operation, and it is not clear how it might be implemented. If anything, this paper *motivates* further investigation of bio-plausible implementations of weight orthogonalization, and experimental investigation into whether weights are indeed close to orthogonal. But these are not settled questions and the paper shouldn't treat them as such.


Review 2

Summary and Contributions: This paper explores the relationship between the backprop learning algorithm and "target prop" or "deep target" algorithms in which each layer is given a target output rather than a gradient. The main contribution of this paper is a clear mathematical analyis showing the non-trivial relationship between backpropagation and target prop in an "idealized" setting of invertible networks (invertible activation functions and invertible weight matrices). While this assumption itself is not biologically plausible, the authors rightly point out that such inverse functions can be approximated using autoencoders as originally proposed in the 2014 paper "Target Propagation" by Lee et. al. The authors use their new understanding to propose and experiment with a particular target prop algorithm (GAIT-prop) that makes this connection between the backprop and targetprop more concrete.

Strengths: Target prop algorithms are of interest in the study of biologically-plausible learning algorithms, and I think this work makes a useful contribution in understanding them better.

Weaknesses: The experiments here highlight the agreement between GAIT-prop and backprop, but do not show that GAIT-prop can be used to train deep networks. In the 4-layer models presented here, one expects to get good performance simply by randomly initializing the weights and training the top layer.

Correctness: In line 147, it is a little misleading to say that this is as biologically plausible as the classic target prop algorithm. There are many different kinds of target prop algorithms, and it is the ones that address the weight transport problem that are considered biologically plausible. The formulation here does not (but could be approximated in a way that does).

Clarity: Yes, the paper was clearly written. I thought the authors presented their mathematical analysis well.

Relation to Prior Work: Yes.

Reproducibility: Yes

Additional Feedback: Equation 4 should be y_{l-1}^T not y_l^T. Equation 17 should be g_{l+1}^{-1} instead of g_l^{-1}. Line 156: This statement about weak coupling and co-existence was unclear to me.


Review 3

Summary and Contributions: This paper is in the field of biologically plausible alternatives to backprop, and introduces a novel variant of target propagation which eliminates some of the problems with earlier approaches (which meant they basically did not work well except in limited settings). The main novelty is with a particular propagation rule of the targets which keeps them close to the feedforward values, with a convex weight chosen to make the resulting update match the true gradient when the weight matrices are square and orthogonal, up to a per-layer scaling factor. Experiments suggest that this approach perfectly tracks backprop when the true inverse of the forward propagation of each layer is used in the backward phase, on MNIST, KMNIST and Fashion-MNIST.

Strengths: The proposed approach is theoretically justified (also see the scaling problem below) and seems to work much better than earlier attempts in small-scale experiments presented in the paper. This is a serious achievement, given that many researchers have tried to make this idea work over the last few years. An interesting feature of the proposed approach (not noted by the authors) is that inputs and targets don't have to be provided with a precise timing difference (they could even be provided together, it is just that gradient calculations will become valid only after both a full forward pass and a full backward pass have been completed).

Weaknesses: There are still many outstanding issues to approach biological plausibility (but there is no work which has achieved perfect biological plausibility, so this is more a list of future things to do in order to follow-up on this paper): - use an approximate inverse (e.g. using some form of auto-encoder), as suggested in the paper and done in earlier work - actually use a biologically plausible regularizer to achieve approximate orthogonality, in the experiments - the learning rates must be scaled exponentially as a function of layer size and this is completely implausible, this is probably the most serious problem - some external clocking mechanism is still required to know when gradient calculations has completed and updates can proceed (or some other mechanism to avoid that must be devised). - real neural networks have feedback on the same neurons used in the feedforward phase, not matching the separate paths proposed here.

Correctness: The claims and the methodology seem correct.

Clarity: Very well. It should have been said earlier that g^{-1} was computed exactly rather than by some approximation.

Relation to Prior Work: Very good. Note that LeCun's thesis included a derivation of backprop which also has the flavour of targetprop with small increments, so it should definitely be cited here.

Reproducibility: Yes

Additional Feedback: Excellent work! Thanks for the comments in the rebuttal. It would be good to include comparisons with feedback alignment variants, and finally to scale this to convnets, larger nets, and larger datasets like ImageNet. I don't see any computational reasons for not doing it. This would be an even more impressive result.


Review 4

Summary and Contributions: This paper studies the link between two existing algorithms: backpropagation (BP) and target propagation (TP). The authors introduce a variant of TP called “gradient-adjusted incremental target propagation” (GAIT-prop). In GAIT-prop, the target is a small perturbation of the forward pass. They derive an exact correspondence between GAIT-prop and BP in a multi-layer perceptron. This line of work is related to the NGRAD hypothesis by Lillicrap et al (neural gradient representation by activity differences’), i.e. the hypothesis that error gradients are encoded in the brain as differences of neural activities.

Strengths: The authors show mathematically an equivalence between their algorithm (GAITprop) and backpropagation, when synaptic weight matrices are orthogonal. Under this assumption they show that backpropagation and GAIT-prop give identical updates.

Weaknesses: My major concern is that, in algorithm 1, it is not clear to me how computing g^-1 can be done in biological brains and/or in neuromorphic hardware. Another limitation of this approach is that it seems to require many assumptions. In particular, the weight matrices must be square. Also, the GAIT-prop algorithm is presented only in the case of regular multi-layered networks. Does this approach generalize beyond this simple setting to more ada architectures too? What happens if there are skip-layer synaptic connections for example? Another problem is about the orthogonality assumption: even if orthogonality holds at a particular point in time, after only one weight update this assumption will break (unless conditions are enforced to guarantee orthogonality throughout training?).

Correctness: Beware that some statements in the paper reflect more your personal opinion than actual scientific truths. See below for suggestions for improvement.

Clarity: I found the mathematical notations and equations somewhat difficult to follow. See below for suggestions for improvement

Relation to Prior Work: The relation to target propagation is well explained. The authors also very briefly mention a connection to the recirculation algorithm and the equilibrium propagation algorithm without details.

Reproducibility: Yes

Additional Feedback: ==== Post rebuttal ==== Thanks for the clarifications in the rebuttal. I have revised my review and increased my score to 6. Maybe I am wrong, but I wonder if it is necessary to have gamma_1, ... gamma_{L-1} tend to 0 in your theoretical result (?) As gamma_L tends to 0 (with gamma_1, ... gamma_{L-1} being fixed), every layer-wise target t^dagger_k tends to the layer's activity y_k. I would expect that this is all is needed to show the equivalence, but maybe I am wrong. (In fact, I wonder if gamma_1, ... gamma_{L-1} are useful at all in the result you show) Here is a list of suggestions to improve clarity of the paper. “However, it is now clear that purely Hebbian learning rules are not effective at learning complex behavioral tasks.” I would recommend to reformulate this sentence in such a way that shows that this is only the authors' opinion, not a well established fact. Line 116-117 "assuming g_L(·) to be a continuous function" Here I guess that you mean "continuously differentiable", not continuous. In the pseudo code (algorithm 1 page 5), you define \ell_l(W_l) and then write "Update W_l by SGD on the quadratic loss function \ell_l(W_l)". I suppose here that only y_l is seen as depending on W_l, and that the target t_l^\dagger is seen as a constant (otherwise we'd have to differentiate also through t_l^\dagger wrt W_l). The identity matrix is sometimes denoted by I (e.g. line 106) and sometimes by 1 (e.g. Eq17). Eq 12 in section 4. By first reading this, I found it difficult to understand why you scale each target t_{l-1}* by a factor gamma_{l-1}. It's only in section 5 that things become more clear why you do that. But then it's not clear to me if it's really necessary to do this layer-wise scaling (see my question above). In Eq20, the products of matrices are written as if everything was commutative, but products of matrices are not commutative in general. In this case, the first product must be understood as a product from j=l+1 to j=L, and the second one as a product from j=L down to j=l+1. There seems to be a mistake in Equation 4. I suppose this should be y_{l-1}^T. In Equation 3, the terms corresponding to layer L and layer l-1 are next to each other, which makes it a bit difficult to read at first. Lines 21-22: “Deep networks have been shown to replicate the hierarchy of cortical representations“ The term “replicate” seems too strong here. To my knowledge, the references cited show that, in some sense, cortical representations better correlate with deep network models (AlexNet in this case) than with other models proposed by neuroscientists. Line 83: “relatively high performance”. Could be more specific The notation epsilon is used in line 116 and in Eq16 to mean two different things. Also, in the pseudo code (algorithm 1), the index to epsilon seems to be missing. “we show that TP and BP have the same local optima in a deep linear network.” Technically, an optimum refers to a point which minimizes a loss (a scalar function). Here TP and BP refer to learning rules, not loss functions. (there are examples of learning rules that never converge, e.g. CD-1 in some pathological case). Typos: line 123: "obtain obtain", line 224: 24 * 24 instead of 28 * 28, line 226: "minimisd"

[Author Response · NeurIPS 2020]

Please allow us to start by thanking the reviewers and area chairs for their consideration and time. Before providing rebuttals to specific concerns, we would like to emphasize a major contribution of this work, which was perhaps overlooked by some of the reviewers. Specifically, **a core strength of this work is our theoretical demonstration of an exact relationship between updates computed by backpropagation of error and those computed by a learning rule using local targets**. The demonstration of such a relationship (and the conditions under which it holds) has eluded researchers thus far, as also articulated by Reviewer 3. Considering the importance of local target-based learning for both biologically plausible learning and neuromorphic chips, this theoretical development is inherently valuable.

Let us now address the concerns of our reviewers. In order to provide an integrated response, we have grouped the major concerns into three main categories: the use of exact inversions, restrictions surrounding orthogonal weight matrices, and finally the depth/sophistication of the models tested.

The use of exact inverse operations, which allow accurate inversion of our targets from output to hidden layers, was a common concern among our reviewers. The biological plausibility of such operations is undeniably questionable and inverse models in the brain would require plausible learning mechanisms (e.g. via an auto-encoder-like training process or an approach such as that described by [1]). However, **our use of exact inverses was motivated by two main desires: a clear theoretical exposition, and a robust comparison against the baseline (target propagation)**. Exact inverses allowed us to theoretically develop the aforementioned relationship between backpropagation and GAIT-propagation. Furthermore, we could isolate the performance differences between the traditional target propagation and GAIT-propagation methods, establishing an upper bound of performance without the requirement for expensive parameter tuning of an auto-encoder for the inverse model (just as the authors of [2] make use of perfectly symmetric weight matrices when testing the ideal conditions of their biologically plausible model). Without this rigid structure, comparisons of these methods (both theoretically and empirically) would have been made opaque.

Justifications of our approach aside, we agree that, ultimately, the methods we described ought to be recreated with entirely biologically grounded components for maximum impact and utility in the computational neuroscience community. Such a biologically grounded implementation would include plausible and effective implementations of learned inverses, mechanisms for weight matrix orthogonalization, and online learning. We consider our current work to provide an ideal theoretical basis suitable for the NeurIPS community and are excited as authors to explore biologically motivated extensions in the future.

Our discussion of plausibility naturally leads us to another criticism (see reviews 1 and 4): orthogonal weight matrix requirements for GAIT-prop and their enforcement. On this account we could have been more explicit regarding the empirical observations made of GAIT-prop. **We observe empirically that a weak orthogonality regularizer is sufficient for high performance of the GAIT-prop training approach.** Our submitted Supplementary Material can be examined in order to confirm that our parameter sweeps show the stabilization of network training and achievement of competitive performance with GAIT-prop when using a relatively small orthogonality regularizer. We would therefore suppose that known decorrelation mechanisms of the brain would be sufficient for good training performance in practice. Decorrelation could be integrated in future studies by taking inspiration from existing work in computational neuroscience literature [3]. Furthermore, we observe that even without orthogonal regularization/orthogonalization shallow networks and networks with reducing width can successfully be trained (see green lines labelled "GAIT-prop" in Figures 2 and 3 of our submission). We believe that this resolves the reviewer criticism on this point.

Finally, Reviewers 2 and 4 highlighted limitations in the complexity of our datasets and depths of the networks we trained. On the account of model depth, we wish to draw attention to our Supplementary Material in which **we showed robust training of networks by GAIT-propagation up to depths of 6 and 8 layers.** We would expect that in much deeper networks, and without strictly orthogonal weight matrices, there would be impaired learning and we are happy to make this clearer in the text. However given the neuro-centric approach taken here, we must also consider that extremely deep feedforward neural networks do not reflect the structure of biological neural networks. For example, visual processing occurring between 50 and 100ms of stimulus presentation is largely feed-forward but involves a limited number of cortical 'layers'. Reviewer 4 also mentioned the extension of our approach to alternative architectures. On this note we are excited to see further theoretical developments in this direction, however our aim in this paper was to reconcile current developments in target-based learning, which are largely explored in feed-forward neural networks.

[1] Mohamed Akrout, Collin Wilson, Peter C Humphreys, Timothy Lillicrap, and Douglas Tweed. Deep learning without weight transport. *ArXiv*, 1904.05391, April 2019.

[2] Alexandre Payeur, Jordan Guerguiev, Friedemann Zenke, Blake A Richards, and Richard Naud. Burst-dependent synaptic plasticity can coordinate learning in hierarchical circuits. *bioRxiv*, 2020.03.30.015511, March 2020.

[3] Paul D King, Joel Zylberberg, and Michael R DeWeese. Inhibitory interneurons decorrelate excitatory cells to drive sparse code formation in a spiking model of V1. *J. Neurosci.*, 33(13):5475–5485, March 2013.


[Meta-Review · NeurIPS 2020]

This paper presents a biologically plausible learning rule as an alternative to standard back-propagation. This is a heavily studied area in ML, with strong interest from both the ML and computational neuroscience communities. The reviewers agreed that this work presents an exciting and important contribution over the existing literature on this problem. There was extensive discussion between reviewers, with two reviewers championing the paper for acceptance. The lower scoring reviewers cited the empirical evaluation as a weakness of the paper, while others argued that the idea on its own was sufficiently interesting to the community. Ultimately, all reviewers agreed that this paper should be accepted and should generate interesting discussion at the conference.